# Genetic Predisposition and Disease Expression of Bipolar Disorder Reflected in Shape Changes of the Anterior Limbic Network

**DOI:** 10.3390/brainsci9090240

**Published:** 2019-09-19

**Authors:** Chia-Feng Lu, Yu-Te Wu, Shin Teng, Po-Shan Wang, Pei-Chi Tu, Tung-Ping Su, Chi-Wen Jao, Cheng-Ta Li

**Affiliations:** 1Department of Biomedical Imaging and Radiological Sciences, National Yang-Ming University, Taipei 11221, Taiwan, China; ytwu@ym.edu.tw (Y.-T.W.); iamtengshin@gmail.com (S.T.); 2Institute of Biophotonics, National Yang-Ming University, Taipei 11221, Taiwan, China; b8001071@yahoo.com.tw (P.-S.W.); c3665810@ms24.hinet.net (C.-W.J.); 3Brain Research Center, National Yang-Ming University, Taipei 11221, Taiwan, China; 4The Neurological Institute, Taipei Municipal Gan-Dau Hospital, Taipei 11260, Taiwan, China; 5Department of Neurology, Taipei Veterans General Hospital, Taipei 11217, Taiwan, China; 6Department of Psychiatry, Taipei Veterans General Hospital, Taipei 11217, Taiwan, China; peichitu@gmail.com (P.-C.T.); tomsu0402@gmail.com (T.-P.S.); 7Department of Medical Education and Research, Taipei Veterans General Hospital, Taipei 11217, Taiwan, China; 8Division of Psychiatry, Faculty of Medicine, National Yang-Ming University, Taipei 11221, Taiwan, China; 9Department of Psychiatry, Cheng-Hsin General Hospital, Taipei 11220, Taiwan, China; 10Institute of Brain Science, National Yang-Ming University, Taipei 11221, Taiwan, China

**Keywords:** bipolar disorder, hereditary predisposition, cortical folding structures, anterior limbic network, shape index, curvedness

## Abstract

Bipolar disorder (BD) is a genetically and phenotypically complex psychiatric disease. Although previous studies have suggested that the relatives of BD patients have an increased risk of experiencing affective disturbances, most relatives who have similar genotypes may not manifest the disorder. We aim to identify the neuroimaging alterations—specifically, the cortical folding structures of the anterior limbic network (ALN)—in BD patients and their siblings, compared to healthy controls. The shared alterations in patients and their siblings may indicate the hereditary predisposition of BD, and the altered cortical structures unique to BD patients may be a probe of BD expression. High-resolution, T1-weighted magnetic resonance images for 17 euthymic patients with BD, 17 unaffected siblings of BD patients, and 22 healthy controls were acquired. We categorized the cortical regions within the ALN into sulcal and gyral areas, based on the shape index, followed by the measurement of the folding degree, using the curvedness. Our results revealed that the changes in cortical folding in the orbitofrontal and temporal regions were associated with a hereditary predisposition to BD. Cortical folding structures in multiple regions of the ALN, particularly in the striatal–thalamic circuit and anterior cingulate cortex, could be used to differentiate BD patients from healthy controls and unaffected siblings. We concluded that the cortical folding structures of ALN can provide potential biomarkers for clinical diagnosis of BD and differentiation from the unaffected siblings.

## 1. Introduction

Bipolar disorder (BD) is a genetically and phenotypically complex psychiatric disease [1,2]. Previous studies have suggested that the unaffected relatives of patients with BD have an increased risk of experiencing affective disturbances than relatives of healthy controls [2,3,4]. Although genetic factors strongly influence the development of BD [5,6,7], most relatives of patients with BD (approximately 60%) who have similar genotypes will not manifest the disorder [8]. An alternative method of examining disease expression of BD is comparing the phenotypes unique to BD patients with those of unaffected relatives and healthy controls. The shared alterations present in both patients with BD and their unaffected relatives, but not present in healthy controls, could be associated with a hereditary predisposition to BD.

Neuroimaging studies based on magnetic resonance imaging (MRI) have been widely used to reveal changes in brain structure associated with disease expression and hereditary predisposition to BD. Meta-analysis studies of structural MRI have demonstrated that reduced volumes of the prefrontal lobe, anterior cingulate cortex, insula, and corpus callosum, as well as increased volumes in the globus pallidus and the lateral ventricles were associated with BD disease expression [9,10,11,12]. Recent studies have further reported that unaffected relatives may have similar changes in brain structure as BD patients. Changes in brain structure associated with hereditary predisposition to BD include changes in the volume of gray matter (GM) in the right inferior frontal gyrus, anterior cingulate cortex, left insula, and striatum [13,14,15,16,17,18,19]. Relatives genetically likely to have BD exhibited small white matter (WM) volumes and impaired WM connectivity in the frontal, temporal–parietal, and parietal regions [14,17,20,21]. The findings of these volumetric studies suggest that changes in the regions of the anterior limbic network (ALN)—i.e., the prefrontal, medial temporal, and subcortical regions—can be a potential marker for disease expression and hereditary predisposition to BD. However, in the majority of previous studies, researchers have detected local changes in the concentration of neuronal cells or cortical volume by using voxel-based morphometric or cortical thickness analysis. The macroscopic alterations in cortical folding structures associated with disease expression and hereditary predisposition to BD have been less explored.

The cortical folding structure is considered to be the substrate for mental state, intelligence, and cognitive functions that distinguish humans from other mammalian species [22,23]. Although the mechanism of cortical folding is unclear, previous studies have provided several hypotheses to explain cortical folding, including a consequence of lateral expansion of GM within the restricted space of the skull [24], tension of WM that connects various cortical regions [25], and the relation between the thickness of the supragranular (i.e., cortical layers I–III) and infragranular (i.e., cortical layers V–VI) layers [23]. Previous research on changes in cortical volume, declines in WM microstructures, and fiber integrity [14,15,16,17,18,19,20,21] have indicated that the cortical folding structures of patients with BD and their unaffected relatives may be disrupted. Several reports have proposed altered folding structures in the prefrontal cortex of patients with BD, measured by the gyrification index or sulcal index [26,27]. Researchers have also observed several cognitive deficits, such as deficits in response inhibition, executive control, verbal memory, and sustained attention, in families that are hereditarily predisposed to BD [28,29,30,31,32]. These findings suggest that the affected cortical structures may be related to the ALN [33]. In this study, we hypothesize that changes in cortical folding structures involved in the ALN could reflect disease expression and hereditary predisposition to BD.

To confirm our hypothesis, we investigated the differences in cortical folding structures between patients with BD, their unaffected siblings, and healthy controls, by using shape index (SI) as well as curvedness (CVD), which is a surfaced-based morphometric method [34,35,36,37]. We have applied this approach to examine fetal brain development [34], and differentiate patients with remitting depression from those with non-remitting depression before medication treatment [36]. The purpose of this study was to determine the following: (1) the brain regions associated with hereditary predisposition to BD, based on the changes in cortical folding structures occurring in patients with BD and their unaffected relatives, but not in healthy controls; (2) the brain regions associated with disease expression of BD, based on the changes in cortical folding structures occurring in patients with BD, but not in unaffected relatives or healthy controls; and (3) the brain regions associated with an absence of BD diagnosis, based on the changes in cortical folding structures occurring in unaffected relatives, but not in patients with BD or healthy controls.

## 2. Materials and Methods

### 2.1. Participants

The Institutional Review Board of Taipei Veterans General Hospital approved this study (protocol No. V100E6-002). Overall, 56 participants were recruited with written informed consent, including 17 euthymic patients with bipolar I disorder, 17 unaffected siblings of patients with BD, and 22 healthy controls from Taipei Veterans General Hospital. Clinical psychiatrists confirmed the BD diagnosis in the 17 patients in the BD group (mean age: 42.9 years; 11 males), according to the fourth edition of the Diagnostic and Statistical Manual of Mental Disorders (DSM-IV), and confirmed that the patients had no comorbidities, including schizophrenia, obsessive-compulsive spectrum disorders, and post-traumatic stress disorder. All patients with BD met the criteria for euthymic state, which includes a score of less than 7 on both the Hamilton Depression Rating Scale and Young Mania Rating Scale on the day of imaging. Medication that patients with BD used before their participation in this study included mood stabilizers, atypical antipsychotics, and hypnotic medications. The 17 unaffected siblings (mean age: 42.6 years; 8 males), as well as 22 age- and gender-matched healthy controls (mean age: 42.1 years; 13 males), all underwent the same diagnostic evaluation to confirm the absence of psychiatric illness. All participants were right-handed and had no history of neurological or physical disorders, alcohol or drug dependence, or electroconvulsive therapy. Table 1 lists the demographic and clinical details of all the participants.

### 2.2. Imaging Data Acquisition and Preprocessing

Three-dimensional, T1-weighted images for each participant were acquired at Taipei Veterans General Hospital by using a Discovery MR750 3T system (GE Healthcare, Chicago, IL, USA), with a rapid acquisition gradient echo. All the participants were instructed to close their eyes, hold still, and relax during the acquisition. The image parameters were repitition time (TR) = 12.2 ms, echo time (TE) = 5.2 ms, flip angle = 12°, field of view (FOV) = 256 × 256 × 168 mm^3^, matrix size = 256 × 256 × 168, slice thickness = 1 mm, and voxel size = 1 × 1 × 1 mm^3^.

The skull-stripping and intensity non-uniformity correction on the T1-weighted images were performed using a hybrid watershed algorithm [38], as well as nonparametric, nonuniform intensity normalization [39] implemented by the Freesurfer software (http://surfer.nmr.mgh.harvard.edu). The brain tissue was then further segmented into GM, WM, and cerebrospinal fluid (CSF) areas by using the Statistical Parametric Mapping 12 package (http://www.fil.ion.ucl.ac.uk/spm).

### 2.3. Cortical Surface Extraction and Shape Analysis

We extracted the GM–WM boundary (i.e., the inner cortical surface), in order to estimate the cortical morphology. The reason that we extracted the inner cortical surface, rather than the outer cortical surface (GM–CSF boundary), was to decrease partial volume errors, which occur frequently on the highly folded outer surface. Additionally, the outer cortical surface contour is difficult to precisely identify when the brain becomes increasingly folded, forming the classical “T” shape [40]. Therefore, we applied cortical shape analysis on the inner cortical surface, in order to obtain consistent and reliable calculations.

Regarding the shape analysis of the cortical folding structure, shape index (SI) and curvedness (CVD) are derived from the polar system of principal curvatures, which includes the maximal (*k*_1_) and minimal curvature (*k*_2_) [41]. We used the DIPimage toolbox (http://www.diplib.org) to calculate the principal curvatures directly from the binary volume image (i.e., we used the one and zero intensity to represent the volume of WM and other background voxels, respectively), without reconstructing the surface [42]. The SI and CVD from various combinations of *k*_1_ and *k*_2_ were computed as follows:(1)$$\mathrm{SI}=\frac{2}{\pi }arctan\frac{k_{2}+k_{1}}{k_{2}-k_{1}}\left(k_{2}\geq k_{1}\right)$$
(2)$$\mathrm{CVD}={\left(\frac{k_{1}^{2}+k_{2}^{2}}{2}\right)}^{1/2}$$

The SI values range between −1 and 1, whereas the CVD values are always positive, representing the extent of the folding structure (Figure 1).

The surface areas were further separated into two categories—namely, the gyral surface (an outward structure with SI > 0) and the sulcal surface (an inward structure with SI < 0; Figure 1). The separation between the gyral and sulcal surfaces can effectively evaluate morphological changes in brain development [34] and degeneration [36]. To investigate the cortical shape changes of the ALN, we calculated the mean CVD values of the gyral and sulcal surfaces separately in the orbital frontal cortex, anterior temporal cortex, anterior cingulate cortex, hippocampus, parahippocampal gyrus, amygdala, thalamus, and striatum [43] (covering 14 regions of the Anatomical Automatic Labeling atlas [44]) in each hemisphere.

### 2.4. Statistical Analysis for Group Comparison

We examined the group differences in the mean CVD values of cortical regions within the ALN using one-way analysis of variance statistics (*p* < 0.05). We conducted a post-hoc, pairwise *t*-tests (patients with BD vs. healthy controls, patients with BD vs. unaffected siblings, and unaffected siblings vs. healthy controls) with Bonferroni correction for multiple comparisons.

## 3. Results

### 3.1. Changes in the Cortical Folding associated with Hereditary Predisposition to Bipolar Disorder

The multiple comparison tests (*p* < 0.05 with Bonferroni correction) demonstrated that both the patients with BD and unaffected siblings exhibited significantly lower CVD values in the orbitofrontal and temporal regions than did the healthy controls (Figure 2a, Table 2, and Figure 3a). Specifically, the CVD values of the patients with BD and unaffected siblings decreased significantly on the right superior and middle orbitofrontal gyral surfaces, as well as on the right superior-medial orbitofrontal sulcal surface. In the temporal region, we observed significantly lower CVD values on the left inferior temporal gyral surface in BD patients and unaffected siblings, compared to the healthy controls.

We also observed that the patients with BD and unaffected siblings exhibited decreased CVD values in the same brain region, but in different surface categories. In the left insula, we observed significantly low CVD values on the sulcal surfaces of the patients with BD, but on the gyral surfaces of the unaffected siblings.

### 3.2. Changes in Cortical Folding associated with Disease Expression of BD

The patients with BD also exhibited significantly lower CVD values in the several regions of the ALN than did the healthy controls (Figure 2b, Table 3, and Figure 3b). Specifically, patients with BD exhibited disruption in the cortical folding structures of the right insula, bilateral hippocampus, right caudate, right thalamus, bilateral superior temporal poles, and right inferior temporal gyrus. The majority of changes in the cortical folding structures of patients with BD occurred on the sulcal surfaces. However, the CVD values of these regions did not statistically differ from those of the unaffected siblings.

We observed that the CVD value of the sulcal surface of the anterior cingulate cortex of patients with BD was significantly lower than that of the unaffected siblings.

### 3.3. Changes in Cortical Folding Associated with the Absence of Bipolar Disorder Diagnosis

We observed significant cortical folding changes uniquely associated with the absence of a clinical diagnosis of BD in the unaffected siblings in the right caudate gyral surfaces (Figure 2c, Table 4, and Figure 3c). The unaffected siblings exhibited lower CVD values in these two regions than did the healthy controls and the patients with BD.

Compared with the healthy controls, the unaffected siblings exhibited significantly lower CVD values on several gyral surfaces of the ALN, including the right parahippocampus, left amygdala, and right putamen.

## 4. Discussion

This study investigated the differences in the cortical folding structures of ALN between patients with BD, unaffected siblings, and healthy controls, by using combined SI and CVD analysis. We found that cortical folding changes in the orbitofrontal and temporal regions were associated with a hereditary predisposition to BD. Furthermore, disease expression of BD was associated with the disruption of folding structures in the striatal–thalamic circuit and the anterior cingulate cortex, compared to the healthy controls and unaffected siblings.

### 4.1. Neurobiological Basis of Changes in Cortical Folding Structure

Previous studies have reported the BD-related changes in cortical folding structures [45]. McIntoch et al. demonstrated that patients with BD exhibited reduced folding complexity in the prefrontal cortex, by using a gyrification index to determine the degree of gyral convolution [26]. Penttilä et al. discovered decreased sulcal patterns in the right prefrontal regions of patients with BD by using a sulcal index, which was the ratio between the sulcal area and total outer cortex area [27,46]. Our findings supplement previous studies on folding changes in patients with BD, to reveal associations between cortical folding structures, hereditary predisposition, and disease expression of BD. Our results also indicate that disease expression of BD could disrupt the cortical folding structures within the ALN. Previous studies have suggested that genetic and environmental factors cause variations in the fetal-to-childhood folding formation process [22,47,48]. The results of our research on both patients with BD and unaffected siblings confirm that the hereditary predisposition and familial risk for BD could alter the cortical folding patterns of ALN.

Measuring cortical folding structures provides macroscopic information on the underlying architectures of the brains of patients with BD and unaffected relatives, which cannot be obtained by examining local GM and WM changes alone. Although several hypotheses have proposed explanations for the mechanisms underlying cortex folding, previous studies have suggested that folding patterns may be influenced by multiple factors [22]. Mota et al. proposed that cortical folding is driven by WM connectivity, and is affected by the fraction of cortical neurons connected through the WM and the average cross-sectional area of the axons in the WM [24]. Accordingly, disruptions in folding cortices (i.e., reduced CVD or flattened sulcogyral patterns) may be related to impaired WM connectivity [20,21,49], as well as the decreased cortical volume and thickness connected through WM [14,17,20,21].

### 4.2. Changes in Cortical Folding Associated with Hereditary Predisposition to Bipolar Disorder

The patients with BD and the unaffected siblings presented disruptions in the cortical folding structures in the orbitofrontal and temporal regions that were not present in the healthy controls (Figure 2a, Table 2, and Figure 3a). Previous studies on twins and families have reported that the structures of these regions are highly heritable [50,51]. The orbitofrontal and temporal regions are not only the components of the ALN, but also resemble the regions in the default mode network that are involved in internal mentation, as well as in emotional and self-referential processing [52,53]. The abnormality of functional connections within the default mode network was identified in the BD patients, as well as in their unaffected first-degree relatives, based on an functional magnetic resonance imaging (fMRI) study [54]. The decline in cortical folding structures in these regions could result in the functional changes of mental representation associated with hereditary predisposition to BD.

We also observed significant folding changes in the left insula of patients with BD. The insula integrates interoception and sensory afferents to generate subjective emotions, and regulates homeostasis and mental states [55,56]. The volume of the left insula could be an indication of emotional dysregulation related to the hereditary predisposition to BD [16,17]. In this study, we observed flattening on the sulcal surfaces of the left insula of patients with BD, as well as on the gyral surfaces of the left insula of the unaffected siblings. Our results support the finding that changes in surface morphology occur mainly in the sulcal areas in patients with BD [27,46].

### 4.3. Changes in Cortical Folding Associated with Disease Expression of Bipolar Disorder

The patients with BD exhibited more affected ALN regions, including the insula, hippocampus, temporal regions, and the striatal–thalamic circuit, such as the caudate and thalamus (Figure 2b, Table 3, and Figure 3b). The striatum–thalamic circuit is the core of the ALN, and regulates socioemotional behaviors related to changes in a person’s emotions, relations with others, and self-concept [33,43]. The temporal regions involved in emotional regulation and reward processing modulate the striatal and frontal regions [57,58]. The results of our previous fMRI study on the functional connectivity of various regions of the brain demonstrated alterations in the striatal–thalamic connectivity of patients with BD [59]. We observed fewer differences in altered ALN regions between unaffected siblings and healthy controls than between patients with BD and healthy controls. In particular, the striatal and thalamic regions exhibited no significant changes between unaffected siblings and healthy controls. Our results suggest that a higher number of affected ALN regions and the involvement of the striatal–thalamic circuit could be associated with disease expression of BD.

We observed significant folding differences in the right anterior cingulate cortex between patients with BD and the unaffected siblings (Figure 2b, Table 3, and Figure 3b). The anterior cingulate cortex is central to integrating emotional and cognitive information for supporting emotional regulation, attention and executive functions, error detection, and working memory [60,61]. Fornito et al. showed that the morphology of the anterior cingulate cortex is associated with executive functions [62]. Furthermore, Tissier et al. proposed that the sulcal pattern of the anterior cingulate cortex may contribute to the function of inhibitory control [63]. Several studies have reported that abnormalities in the anterior cingulate cortex are associated with the affective and cognitive deficits of patients with BD [64,65,66]. A previous study has also suggested that structural alteration of the anterior cingulate cortex is a risk marker of psychosis, which can appear in patients with bipolar I disorder [67] and their relatives [68]. Fornito et al. demonstrated that high-risk individuals who developed a psychotic episode exhibited thinning of the anterior cingulate cortex [67]. In our BD group, more than half of patients (approximately 64.7%) had previously presented with psychotic symptoms (Table 1), resulting in significant changes in the cortical morphology of the anterior cingulate cortex. Based on our results, disruptions in the folding structure of the anterior cingulate cortex in family members hereditarily predisposed to BD could indicate psychotic and cognitive symptoms associated with disease expression of BD.

### 4.4. Cortical Folding Structures Associated with an Absence of Diagnosis for Unaffected Siblings

The unaffected siblings exhibited significantly decreased CVD in several regions of the ALN, compared with healthy controls (Figure 2c, Table 4, and Figure 3c). These changes in the ALN occurred mainly in the circuit composed of the amygdala, parahippocampus, and subcortical structures. The amygdala, parahippocampus, and striatum are subserved in emotional perception, regulation, expression, and memory [69,70,71]. Several functional studies have indicated that unaffected relatives of patients with BD exhibit hyperactivity of the amygdala and putamen when watching emotional faces [70,72]. Two previous structural studies have reported volumetric changes in the amygdala and hippocampus of unaffected relatives of patients with BD [73,74]. The results of our study support the findings of these studies, and provide information on potential structural substrates associated with facial emotional processing that unaffected relatives of patients with BD might lack.

### 4.5. Further Considerations

Several concerns regarding this study should be mentioned. First, the shared and unique folding changes in the patients with BD and unaffected siblings that did not occur in the healthy controls could be caused by genetic–environmental interactions, rather than a genetic susceptibility. The second concern is that we are unable to assess the actual degree of risk for developing BD in the unaffected siblings of patients with BD. The mean age (42.6 years) of the unaffected siblings in this study was close to the late onset age of BD (40–45 years) [27,75]. Follow-up examination of the folding abnormalities in the younger unaffected siblings could be useful for actually predicting BD onset. The third concern is the medications that the patients with BD used before their participation in this study. Previous studies have demonstrated that the administration of mood stabilizers, such as lithium carbonate and valproic acid, over 2 to 6 months may increase the volume of prefrontal, hippocampal, and cingulate GM volumes, which may cause a neurotrophic effect [76,77]. In our study, two patients with BD took lithium carbonate, and 13 patients took valproic acid. However, the duration of medication use prior to participation in the study was less than 1 month. Therefore, we assumed that the medications may have a minor effect on our results. Finally, this study only included a small cohort of patients with BD and their siblings, due to the difficulty in enrolling participants that matched all the inclusion criteria (i.e., patients with euthymic status and with no comorbidities, including schizophrenia, obsessive-compulsive spectrum disorders, and post-traumatic stress disorder). With these rigorous inclusion criteria, we believe that this study can provide meaningful preliminary results to facilitate subsequent studies on this field. However, because of the limited sample size of this study, we did not apply the correction of multiple comparisons for the brain regions, which typically demands a larger number of participants to obtain a significant change. Accordingly, our exploratory analyses should be replicated on an independent and larger sample size to confirm the findings.

## 5. Conclusions

This study investigated the differences in cortical folding structures for BD patients and their unaffected siblings, compared with healthy controls, by using combined SI and CVD analysis. The folding structures in the orbitofrontal and temporal regions of the ALN are associated with a hereditary predisposition to BD. On the other hand, disease expression of BD is associated with disrupted folding structures in the ALN regions, especially the striatal–thalamic circuit and the anterior cingulate cortex. Our results demonstrate that cortical folding structures can reflect hereditary predisposition and disease expression of BD.

## Figures and Tables

**Figure 1 brainsci-09-00240-f001:**
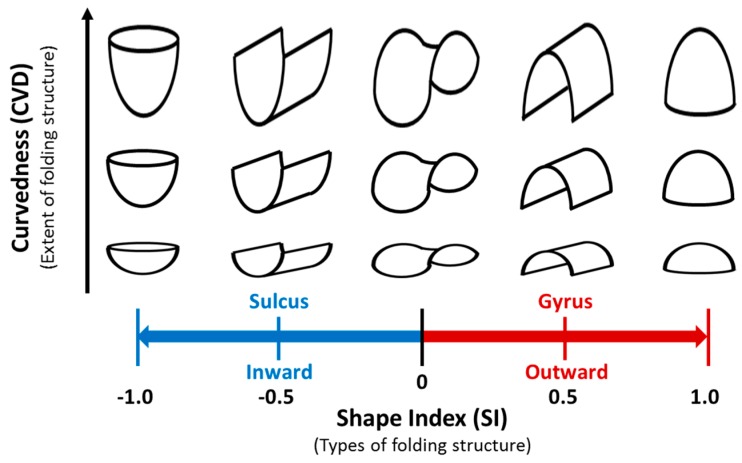
An illustrated diagram of the shape index (SI) and curvedness (CVD). The SI is a measure to describe the types of folding structures, with values ranged from −1 (the inward or sulcal structure) to 1 (the outward or gyral structure); the CVD, on the other hand, measures the extent of a specific folding structure.

**Figure 2 brainsci-09-00240-f002:**
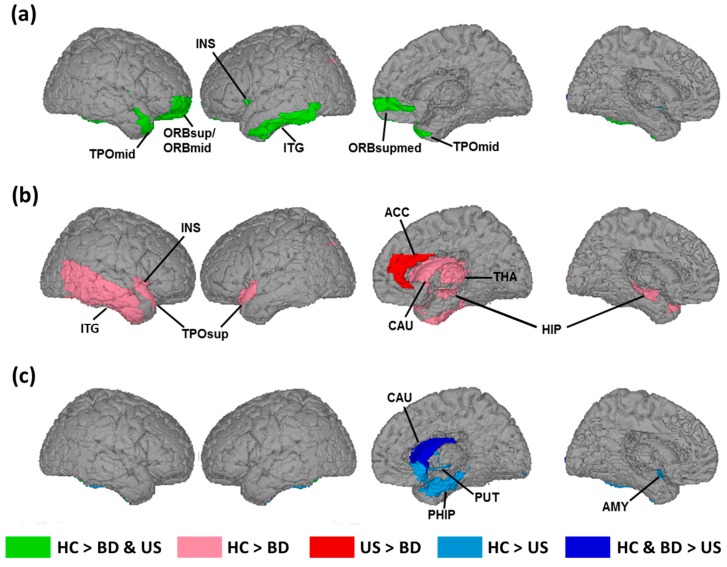
A summarized diagram of brain regions, with significant differences in CVD values between BD patients, unaffected siblings, and healthy controls. (**a**) Significant differences in both BD patients and unaffected siblings, compared with healthy controls; (**b**) significant differences in BD patients compared with healthy controls and unaffected siblings; (**c**) significant differences in unaffected siblings, compared with healthy controls and BD patients. HC: healthy controls; BD: bipolar disorder; US: unaffected siblings; ORBsup: superior orbital frontal gyrus; ORBmid: middle orbital frontal gyrus; ORBsupmed: superior-medial orbital frontal gyrus; TPOsup: superior temporal pole; TPOmid: middle temporal pole; ITG: inferior temporal gyrus; ACC: anterior cingulate cortex; INS: insula; HIP: hippocampus; PHIP: parahippocampal gyrus; CAU: caudate; PUT: putamen; THA: thalamus; AMY: amygdala.

**Figure 3 brainsci-09-00240-f003:**
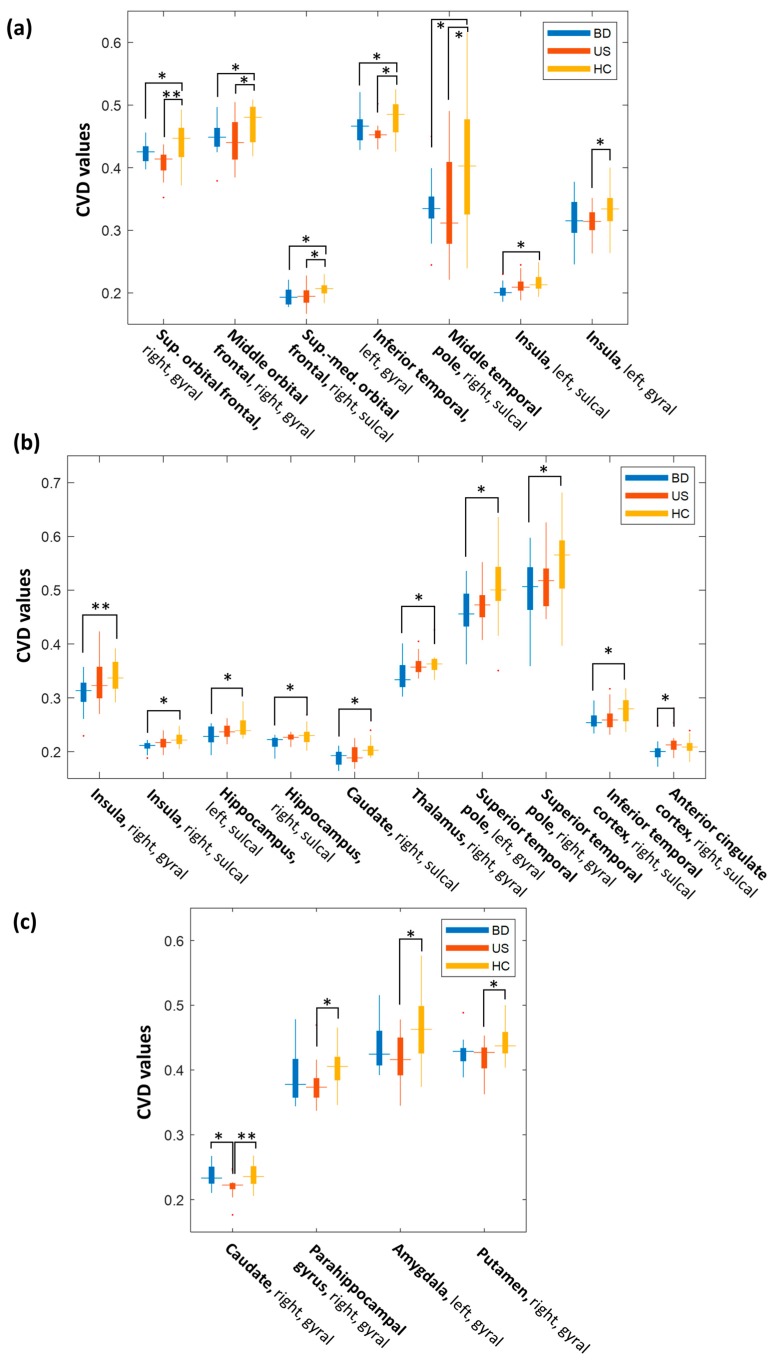
Boxplots for illustrating the group differences in CVD values between BD patients, unaffected siblings, and healthy controls. (**a**) Significant differences in both BD patients and unaffected siblings compared with healthy controls (as listed in Table 2); (**b**) significant differences in BD patients compared with healthy controls and unaffected siblings (as listed in Table 3); (**c**) significant differences in unaffected siblings compared with healthy controls and BD patients (as listed in Table 4).

**Table 1 brainsci-09-00240-t001:** Demographic and clinical details of study cohorts.

	Bipolar I Patients (BD)	Unaffected Siblings (US)	Healthy Controls (HC)	*p*-Value
**Number of subjects**	17	17	22	
**Gender (male/female)**	11/6	8/9	13/9	0.566 ^a^
**Age (years)**	42.9 ± 10.6	42.6 ± 11.2	42.1 ± 10.6	0.970 ^b^
**HAMD**	3.3 ± 2.8	0.7 ± 1.0	1.1 ± 1.4	<0.001 ^b^
**YMARS**	2.4 ± 2.5	0.6 ± 1.8	0.2 ± 0.7	0.001 ^b^
**Duration of illness (years)**	17.3 ± 9.9	--	--	
**Onset age (years)**	25.5 ± 11.5	--	--	
**Past manic episodes (times)**	5.8 ± 6.6	--	--	
**Past depressive episodes (times)**	4.1 ± 3.4	--	--	
**Number of subjects who had psychosis (%)**	11 (64.7%)	--	--	
**GM volumes (cm^3^)**	577.6 ± 56.4	562.3 ± 44.6	569.0 ± 53.8	0.692 ^b^
**WM volumes (cm^3^)**	509.5 ± 56.3	497.8 ± 54.7	506.3 ± 56.8	0.819 ^b^
**GM + WM volumes (cm^3^)**	1087.0 ± 108.4	1060.0 ± 85.6	1075.2 ± 106.1	0.738 ^b^

HAMD: Hamilton Depression Rating Scale, 17 items; YMARS: Young Mania Rating Scale; GM: gray matter; WM: white matter; ^a^ Pearson Chi-Square test; ^b^ One-way analysis of variance (ANOVA).

**Table 2 brainsci-09-00240-t002:** Significant difference of curvedness (CVD) values in both BD patients and unaffected siblings, compared with healthy controls.

Cortical Region	Surface Category	CVD in BD	CVD in US	CVD in HC	Pairwise *p*-Value
***BD < HC and US < HC***					
Superior orbital frontal cortex, right	Gyral	0.424 ± 0.016	0.408 ± 0.022	0.444 ± 0.031	BD vs. HC 0.048 * US vs. HC < 0.001 * BD vs. US 0.228
Middle orbital frontal cortex, right	Gyral	0.448 ± 0.028	0.442 ± 0.037	0.473 ± 0.029	BD vs. HC 0.047 * US vs. HC 0.011 * BD vs. US 1.000
Superior-medial orbital frontal cortex, right	Sulcal	0.194 ± 0.014	0.195 ± 0.014	0.206 ± 0.011	BD vs. HC 0.011 * US vs. HC 0.022 * BD vs. US 1.000
Inferior temporal cortex, left	Gyral	0.463 ± 0.024	0.454 ± 0.016	0.482 ± 0.026	BD vs. HC 0.033 * US vs. HC 0.001 * BD vs. US 0.843
Middle temporal pole, right	Sulcal	0.336 ± 0.047	0.340 ± 0.084	0.405 ± 0.010	BD vs. HC 0.033 * US vs. HC 0.048 * BD vs. US 1.000
Insula, left	Sulcal	0.204 ± 0.011	0.212 ± 0.014	0.215 ± 0.013	BD vs. HC 0.032 * US vs. HC 1.000 BD vs. US 0.197
Insula, left	Gyral	0.319 ± 0.032	0.312 ± 0.024	0.339 ± 0.034	BD vs. HC 0.152 US vs. HC 0.026 * BD vs. US 1.000

US: unaffected siblings; HC: healthy controls. * A significant difference between two groups (*p* < 0.05 after Bonferroni correction).

**Table 3 brainsci-09-00240-t003:** Significant differences in CVD values of BD patients, compared with healthy controls and unaffected siblings.

Cortical Region	Surface Category	CVD in BD	CVD in US	CVD in HC	Pairwise *p*-Value
***BD < HC***					
Insula, right	Gyral	0.308 ± 0.034	0.327 ± 0.041	0.340 ± 0.029	BD vs. HC 0.020 * US vs. HC 0.774 BD vs. US 0.355
Insula, right	Sulcal	0.210 ± 0.009	0.216 ± 0.013	0.223± 0.012	BD vs. HC 0.003 * US vs. HC 0.200 BD vs. US 0.374
Hippocampus, left	Sulcal	0.228 ± 0.019	0.238 ± 0.012	0.244 ± 0.016	BD vs. HC 0.008 * US vs. HC 0.657 BD vs. US 0.243
Hippocampus, right	Sulcal	0.217 ± 0.014	0.226 ± 0.008	0.233 ± 0.022	BD vs. HC 0.011 * US vs. HC 0.574 BD vs. US 0.334
Caudate, right	Sulcal	0.189 ± 0.015	0.194 ± 0.017	0.205 ± 0.014	BD vs. HC 0.006 * US vs. HC 0.068 BD vs. US 1.000
Thalamus, right	Gyral	0.340 ± 0.028	0.361 ± 0.018	0.367 ± 0.030	BD vs. HC 0.009 * US vs. HC 1.000 BD vs. US 0.083
Superior temporal pole, left	Gyral	0.461 ± 0.048	0.484 ± 0.039	0.506 ± 0.070	BD vs. HC 0.045 * US vs. HC 0.251 BD vs. US 1.000
Superior temporal pole, right	Gyral	0.500 ± 0.062	0.515 ± 0.050	0.554 ± 0.066	BD vs. HC 0.022 * US vs. HC 0.152 BD vs. US 1.000
Inferior temporal cortex, right	Sulcal	0.258 ± 0.019	0.263 ± 0.023	0.279 ± 0.026	BD vs. HC 0.018 * US vs. HC 0.104 BD vs. US 1.000
***BD < US***					
Anterior cingulate cortex, right	Sulcal	0.201 ± 0.017	0.216 ± 0.022	0.210 ± 0.013	BD vs. HC 0.262 US vs. HC 0.883 BD vs. US 0.033 *

US: unaffected siblings; HC: healthy controls. * A significant difference between two groups (*p* < 0.05 after Bonferroni correction).

**Table 4 brainsci-09-00240-t004:** Significant differences in CVD values in unaffected siblings, compared with healthy controls and BD patients.

Cortical Region	Surface Category	CVD in BD	CVD in US	CVD in HC	Pairwise *p*-Value
***US < HC and US < BD***					
Caudate, right	Gyral	0.237 ± 0.016	0.220 ± 0.016	0.238 ± 0.017	BD vs. HC 1..000 US vs. HC 0.004 * BD vs. US 0.014 *
***US < HC***					
Parahippocampal gyrus, right	Gyral	0.386 ± 0.036	0.377 ± 0.032	0.405 ± 0.031	BD vs. HC 0.249 US vs. HC 0.036 * BD vs. US 1.000
Amygdala, left	Gyral	0.436 ± 0.036	0.421 ± 0.037	0.461 ± 0.052	BD vs. HC 0.225 US vs. HC 0.016 * BD vs. US 0.918
Putamen, right	Gyral	0.426 ± 0.023	0.419 ± 0.025	0.442 ± 0.024	BD vs. HC 0.136 US vs. HC 0.013 * BD vs. US 1.000

US: unaffected siblings; HC: healthy controls. * A significant difference between two groups (*p* < 0.05 after Bonferroni correction).

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
