# Peer review of "Genetic Predisposition and Disease Expression of Bipolar Disorder Reflected in Shape Changes of the Anterior Limbic Network"

_brainsci, 2019, doi:10.3390/brainsci9090240_

Round 1

Reviewer 1 Report

This an interesting and well-conducted study investigating shape deviations of the Anterior Limbic Network (ALN) in bipolar disorder (BD) patients, unaffected siblings (US) and healthy controls (HC) using an original morphometry approach.

My main concern is the correction of multiple comparisons that is likely applied for the post-hoc comparisons (ie. 3 comparisons: BD vs HC, US vs HC and BD vs US) and not for the comparisons of the different ROIs. If the correction does not apply to the several ROI comparisons, it should then be mentioned that the analyses are exploratory and should then be considered with cautious before replication on an independent, and likely larger, sample.

Minor concerns:

Sulcal alterations in ALN related to BD previously reported should be mentioned in the Introduction Gender and age have a known effect on brain structures. More complex statistical models, including main and interaction effect of gender and age, should therefore be tested. The voxel size should be specified. Maps illustrating SI and CVD should be provided for naïve readers as these metrics are not common. Does CVD correspond to mean curvature. It is not clear why SI is calculated as it is not analyzed in HC, US and BD. Why a ROI-based rather a vertex-wise approach was used ? Plots (e.g. boxplot) to illustrate the group-differences should be provided (it is more easy to interpret plots than Tables with mean and sd). The title of the article is misleading because the reported study does not concern the cortical folding structures of the ALN but the shape of the ALN. Indeed, the Caudate, putamen, thalamus, and amygdala are not cortical regions. The title should then be changed to ‘Genetic Predisposition and Disease Expression of 2 Bipolar Disorder Reflect on *Shape* of the Anterior Limbic Network’. Could deviations reported in ACC curvature be related to impaired cognitive control (e.g. Fornito et al. Cerebral Cortex 2004; Tissier et al. eNeuro 2018) observed in BD? Could differences between US and BD be interpreted as compensatory mechanisms?

Reviewer 2 Report

In this manuscript, authors evaluated the alterations in cortical folding structures of the ALN in three study groups: BD cases, unaffected siblings of BD patients and healthy controls. This is a well organized study with clear description of methods and results. There are few minor issues that authors should address:

1) It is not clear why unaffected relatives do not express BD even though they share similar BD related alterations. The alterations in the cortical folding of BD and US groups and how these alterations might affect the expression of the disease should be discussed in detail.

2) Authors overplay their results in conclusion by making causal assumptions, suggesting folding structures temporal regions of the ALN could be risk markers for hereditary predisposition to BD. Without genetic data, one cannot deduce whether the alterations in the cortical folding is a risk factor (cause) or the consequence of the disorder.

3) There is a typo in Fig1: Figure legend was labeled as a, b, d instead of a, b, c.

Round 2

Reviewer 1 Report

A selection of points raised in my previous review have been addressed, and adequately addressed. However, several main concerns were surprisingly not considered useful to be taken into account.  

I confirm that the following points are important and must be addressed before publication:

Q1.3. Gender and age have a known effect on brain structures. More complex statistical models, including main and interaction effect of gender and age, should therefore be tested.  

>> Even if the three-groups are gender- and age-matched, gender- or age-specific effect may bias the analyses. It is therefore important to investigate possible interaction between group and gender and between group and age.

Q1.9. Plots (e.g. boxplot) to illustrate the group-differences should be provided (it is more easy to interpret plots than Tables with mean and sd).

>> I confirm that a selection of some plots may clearly improve the manuscript.

Q1.11. Could deviations reported in ACC curvature be related to impaired cognitive control (e.g. Fornito et al. Cerebral Cortex 2004; Tissier et al. eNeuro 2018) observed in BD?

>> Even if cognitive data are not available, it is interesting to discuss (and not test) this point.
